energy

HTHP, shear bonding strength, hydraulic bonding strength, casing roughness, temperature variation, internal casing pressure variation

**Author for correspondence:**
Huanqiang Yang
e-mail: yhq840310@sina.com

# Experimental study of shear and hydraulic bonding strength between casing and cement under complex temperature and pressure conditions

Huanqiang Yang[1], Qi Fu[1], Jiang Wu[2], Lulu Qu[1], Dong Xiong[1] and Yang Liu[1]

[1]College of Petroleum Engineering, Yangtze University, Wuhan 430100, People's Republic of China
[2]China National Offshore Oil Corporation (China Limited), Zhanjiang Branch, Guangdong 524057, People's Republic of China

HY, 0000-0001-7490-1870

The cement sheath plays a vital role in preventing gas channelling. It is important to understand the interfacial bonding between the casing and cement sheath when the downhole temperature and pressure change. This paper demonstrates the results of an experimental study to investigate the effect of high temperature and high pressure and their variations on the cement sheath interfacial bonding strength (CSIBS). An experimental device was developed that is used to test the shear and hydraulic bonding strength with the method of uniaxial compression and gas channelling. The results show that both temperature and pressure have a significant influence on the CSIBS. As the curing temperature increases with a constant curing time or as the curing time increases with a constant curing temperature, the CSIBS first increases and then converges to a stable value. The casing roughness has a crucial effect on the shear bonding strength but little effect on the hydraulic bonding strength. Though the CSIBS decreases obviously with the decrease in temperature, it undergoes little change when the temperature first increases and then recovers to the initial value. When the internal casing pressure decreases to a certain value or first increases to a certain value followed by recovery to the initial state, the hydraulic bonding strength tends to be 0 MPa, which means that the interface undergoes debonding between the casing and cement.

# 1. Introduction

After the cementing process, the structure of the oil and gas well consists of casing, cement sheath and rock formation. Formation fluids such as oil and gas are allowed to flow through the inner casing but not through the interfaces between the casing and cement sheath and the cement sheath and rock formation. As an important sealing component, the cement sheath plays a vital role in providing zonal isolation to prevent gas channelling. However, a sustained casing pressure (SCP) between the casing and cement sheath will be generated due to sealing failure in high temperature and high pressure (HTHP) oil and gas wells, which has seriously affected the environment. The long-term sealing integrity of cement sheaths is a prerequisite for ensuring the safe production of oil and gas wells [1–3].

During the lifetime of HTHP oil and gas wells, complex variations in temperature and internal casing pressure generated in the wellbore may lead to sealing failure due to the interfacial debonding between the cement and casing and the cement and rock formation. Sealing failure mechanisms have become an important topic to investigate [4–11]. As the main parameter, the accuracy of cement sheath interfacial bonding strength (CSIBS) is extremely significant for evaluating the sealing integrity of wellbores in HTHP oil and gas wells [12–14]. The CSIBS includes shear bonding strength and hydraulic bonding strength. Shear bonds are essential to support the pipe mechanically, whereas hydraulic bonds prevent the formation of micro-annuli.

However, few studies have been conducted to measure the CSIBS under HTHP conditions. For example, Carter & Evans [15] designed experimental set-ups to measure cement shear bonds and hydraulic bonds between casings and cement, and they demonstrated that the bond properties were both pressure and temperature dependent. However, the temperature considered in the experimental set-ups only increased to 60°C. Goodwin & Crook [16] developed an experimental procedure to study the interfacial failure between casing and cement sheath by testing its permeability by loading/unloading the internal casing pressure, which can acquire the interfacial hydraulic bonding strength. Jackson & Murphey [17] conducted an experiment of bonding strength by improving the cement slurry curing temperature to 49°C and the pressure of the internal casing and annulus to 6.89 MPa. Based on Goodwin & Crook's study [16], Guo et al. [18] investigated the shear bonding strength under the conditions of complicated temperature using the experimental device they developed. However, the experimental temperature only increased to 75°C. Moreover, there are some studies to investigate the shear bond between cement and formation. Gu et al. [19] investigated the effect of interface defects on shear strength and fluid channelling at the cement–interlayer interface. Yang et al. [20] conducted a study concerning the effect of working fluid density on decreasing cement sheath interfacial strength. Xu et al. [21] performed an experimental study concerning the effect of mud cake on the CSIBS, as did Yang et al. [22]. Yang et al. [22] performed an experimental study concerning the shear bonding strength of a cement sheath considering its initial imperfection. Tabatabaei et al. [23] introduced a universal test in conjunction with an analytical solution to measure the mixed mode interfacial strength of cementitious materials at the casing–cement or rock–cement interfaces under room temperature conditions.

Along with oil and gas exploration and development, ultra-deep and offshore oil and gas wells with HTHP conditions have been increasingly developed. As we can see from the above analysis, the high-temperature conditions were not adequately considered, and all experimental processes were conducted without the curing conditions of cement slurry. Moreover, the effect of the surface roughness of the casing on the experimental results was not adequately considered, which greatly affected the accuracy of the test results. Hence, an HTHP interfacial bonding strength device was developed, and the corresponding experiments were carried out.

The structure of this paper is as follows: §2 describes the experimental process, including the experimental device we developed to test the shear and hydraulic bonding strength, the experimental material of the cement slurry system and the shear and hydraulic bonding strength testing method. Section 3 describes the experimental results considering the influencing factors of casing roughness, curing temperature, curing time, temperature variation and internal casing pressure variation.

# 2. Experimental process

## 2.1. Experimental device

Considering the complicated conditions, including long-term high-temperature conditions and variations in temperature and pressure in the casing, an experimental device composed of the following structures was

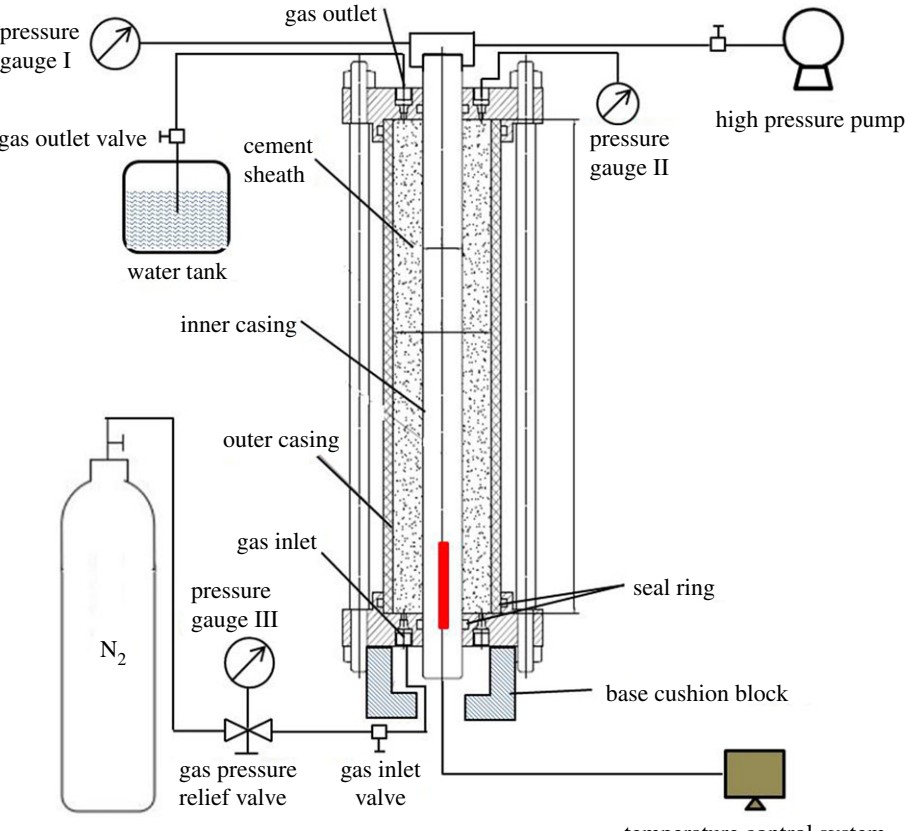

**Figure 1.** Design drawing of the cement sheath interfacial bonding strength testing.

developed, as shown in figure 1. It includes an inner casing, outer casing, cement sheath, gas inlet/outlet, heating rod, temperature control system whose temperature control range is from room temperature to 220°C and is accurate to 0.1°C, and a high-pressure pump that can apply up to 40 MPa. In addition, three pressure gauges (gauge I, gauge II and gauge III) are connected to the valves to separately record the pressure reading of the inner casing, annulus upper cement and injection gas.

## 2.2. Experimental material

In this paper, the high-temperature resistance of a cement slurry system with a density of 2.35 g cm$^{-3}$ used by western South China Sea oilfield is selected as the experimental material. The compositions of the cement slurry system are water, defoamer I (X62L), defoamer II (X66L), dispersant (F41L), fluid loss additive (G80L), latex (GR6), low-temperature high early strength agent (H21L), retarder (H40L), micromax, silica fume (C81) and Portland class G well cement, and the quantities of each component required to prepare a 0.5 l cement slurry are shown in table 1. The rheological property of the cement slurry is tested by an HTHP rheometer, and the rheological parameters are shown in table 2.

## 2.3. Principle and procedure

### 2.3.1. Shear bonding strength

To overcome the limitations of existing experimental test studies of cement sheath shear bonding strength, including the curing temperature of cement, to satisfy demand for HTHP wells and the uncontrollable variation in the temperature and internal casing pressure, the corresponding experimental procedures are designed. First, every agent was weighed by following the composition of the cement slurry system in table 1, and the cement slurry was prepared according to GB/T 33294-2016 'Testing of deepwater well cement'. Second, the prepared cement slurry was poured into the annulus between the inner casing and the outer casing, as shown in figure 2, and the upper part of the cement slurry is filled with water. Third, all the subassemblies of the experimental device are connected, including a high-pressure pipe, high-pressure pump, pressure gauges, heating rod and

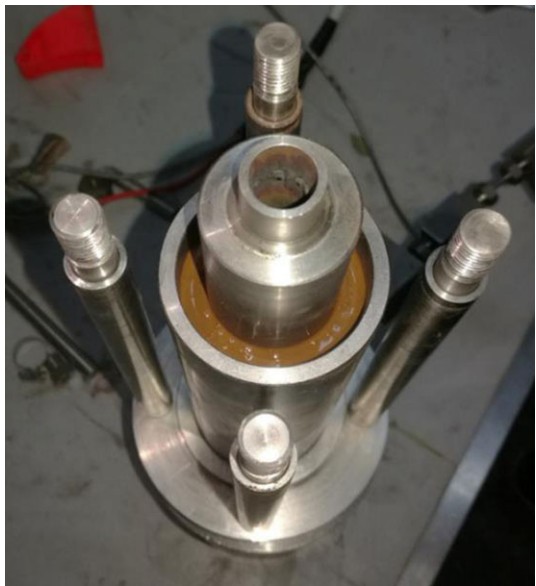

**Figure 2.** Pour the cement slurry into the annulus.

**Table 1.** The composition of the cement slurry system.

| material code | material S.G. | concentration | weight | volume | amount (g) |
|---|---|---|---|---|---|
| water | 1.000 | 4.347 | 38.595 | 38.595 | 164.400 |
| X62L | 1.000 | 0.068 | 0.600 | 0.600 | 2.600 |
| X66L | 1.000 | 0.068 | 0.600 | 0.600 | 2.600 |
| F41L | 1.050 | 0.429 | 4.000 | 3.810 | 17.000 |
| G80L | 1.080 | 0.521 | 5.000 | 4.630 | 21.300 |
| GR6 | 1.040 | 0.542 | 5.000 | 4.808 | 21.300 |
| H21L | 1.240 | 0.091 | 1.000 | 0.806 | 4.300 |
| H40L | 1.180 | 0.095 | 1.000 | 0.847 | 4.300 |
| micromax | 4.850 | 80.000 | 80.000 | 16.495 | 340.800 |
| C81 | 2.630 | 40.000 | 40.000 | 15.209 | 170.400 |
| cement class G | 3.230 | 100.000 | 100.000 | 30.960 | 426.000 |

**Table 2.** The rheological properties of the cement slurry.

| test temperature (°C) | rheological model | liquidity index | consistency coefficient (lbf·s$^n$/100 ft$^2$) |
|---|---|---|---|
| 140 | power law model | 0.72 | 3.13 |

temperature control system. Next, the heat transfer oil was injected into the inner casing using a high-pressure pump to make the heating rod work. During this operation, the inner casing pressure and the annulus pressure will increase with the increase in the system temperature; thus, it is important to control these two pressures through the high-pressure valve connected with the high-pressure pipe. Last, the curing temperature and curing time are set, and the temperature or pressure variation is subsequently determined according to the experimental scheme. The load required is tested with a testing machine to generate a shear slip (as shown in figure 3), which contributes to the measurement of the shear bonding strength:

$$S = \frac{P}{\pi DH},$$ (2.1)

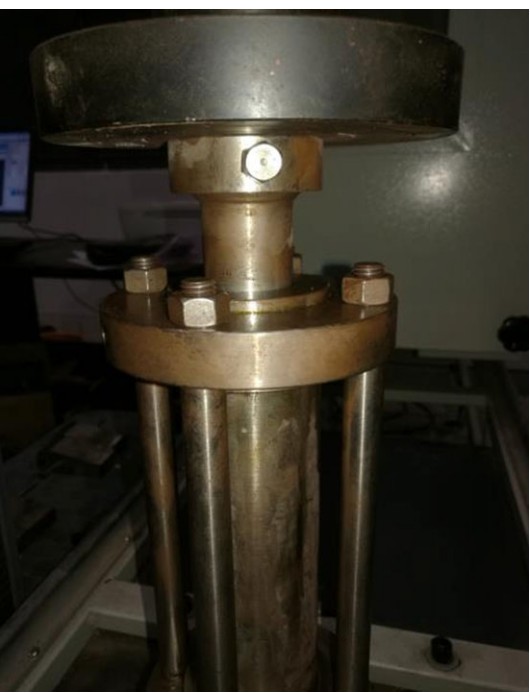

**Figure 3.** Test the shear bonding strength.

where $S$ is the interfacial shear bonding strength, $P$ is the maximum compressive load required to generate the shear slip, and $D$ and $H$ are the inner diameter and the height of the cement sheath, respectively.

### 2.3.2. Hydraulic bonding strength

At present, the cement sheath shear bonding strength is usually adopted by some scholars to represent the interfacial properties. However, the radial debonding between the casing and cement sheath or cement sheath and formation is the main form that leads to seal failure caused by pressure and temperature variations. Therefore, the hydraulic bonding strength is an important parameter to evaluate the integrity of cement sheaths. The cement slurry selected in the study is used for HTHP gas wells in the western South China Sea oilfield. Therefore, the gas channelling method (as shown in figure 4) is used to measure the hydraulic bonding strength under complicated temperature and pressure with the experimental procedures as follows. First, a high magnetic magnet and gauze net at 200 mesh are used to cover the air inlet to prevent the cement slurry from entering during the curing period. Then, the above steps, such as the cement slurry preparation and curing, and all the subassemblies of the experimental device connections are repeated. Last, the gas inlet valve and gas outlet valve are opened, and the gas pressure relief valve is gradually adjusted, which will cause the gas pressure to increase slowly until air bubbles migrate into the water tank. The gas pressure shown in pressure gauge III is the hydraulic bonding strength.

## 3. Results and discussion

With the experimental methods described above, the effects of different casing roughnesses, different curing temperatures at the same time, different curing times with the same temperature, different temperature decreases, different increasing values during a temperature cycle period, different pressure increases in casing and different increasing values in an internal casing pressure cycle period on shear and hydraulic debonding strength are analysed.

### 3.1. The roughness of the simulated casing

The casing roughness is a parameter to measure the microcosmic error of the geometrical shape on the casing surface. It has an important influence on the surface bonding between the casing and cement sheath. Three different simulated casings, including cast iron pipe with a roughness of 0.3 mm, refined

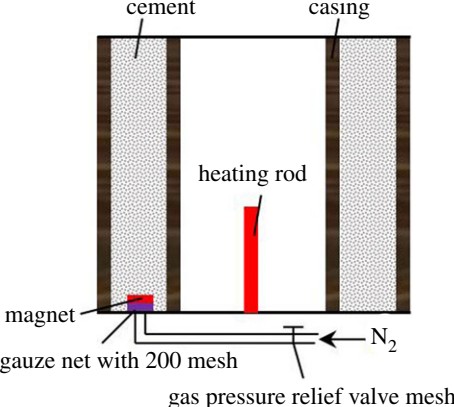

**Figure 4.** The hydraulic bonding strength was tested by the gas channelling method.

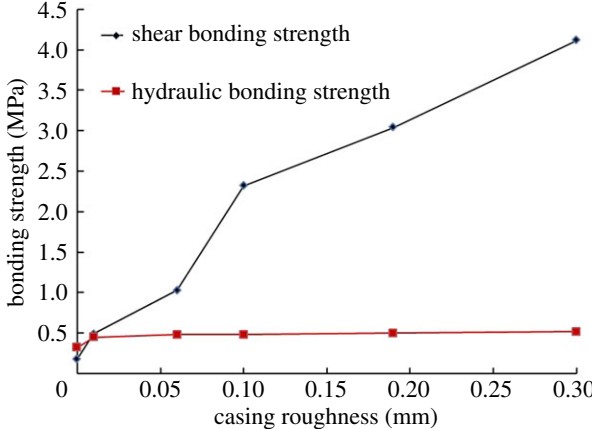

**Figure 5.** The effect of the casing roughness on the bonding strength.

seamless steel with a roughness of 0.06 mm and 0.1 mm, clean seamless steel with a roughness of 0.01 mm and common steel pipe with a roughness of 0.19 mm, are selected in the experiment. Moreover, clean seamless steel whose outer surface is coated with grease is also used to ideally simulate the situation when the roughness is 0 mm. The experimental results of the simulated casings with different roughnesses are shown in figure 5.

Figure 5 shows the change in the shear bonding strength and hydraulic bonding strength with the casing roughness. As the casing roughness increases, the contact area between the casing and cement sheath will increase. Accordingly, the shear bonding strength, which has a constant value per unit area, will increase. By contrast, the hydraulic bonding strength is minorly affected by the contact area. Compared with the casing used in oilfields, the refined seamless steel has a similar roughness used in the following experiments.

## 3.2. Curing temperature

Five different curing temperatures with the same curing time (48 h) are conducted to test the effects of curing temperature on the CSIBS, and the results are shown in figure 6.

Figure 6 indicates the change in the shear and hydraulic bonding strength with increasing temperature for the same curing time. Because of the slow development of cement strength in a low-temperature environment, the shear and hydraulic bonding strength increase 84% and 50%, respectively, as the curing temperature rises from 120°C to 200°C.

## 3.3. Curing time

Due to the strength retrogression of the cement sheath under high-temperature conditions, the CSIBS may decrease accordingly. Therefore, seven different curing times with the same curing temperature (200°C) are considered during the experiment. The results of the CSIBS are shown in figure 7.

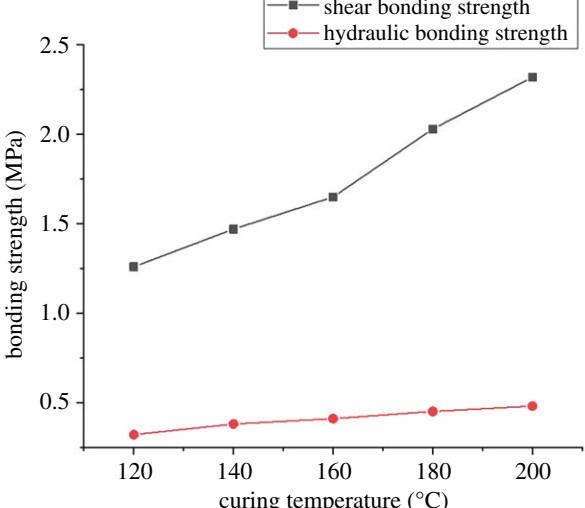

**Figure 6.** The effect of the curing temperature on the bonding strength.

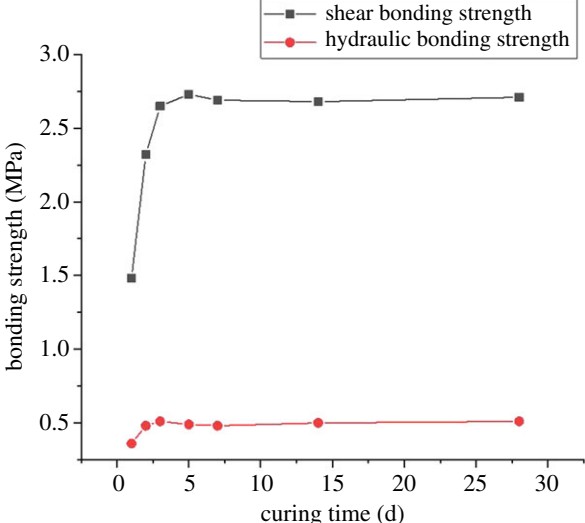

**Figure 7.** The effect of the curing time on the bonding strength.

Figure 7 shows that the shear and hydraulic bonding strength of the cement sheath reveal a marked increase with the same curing temperature as the curing time progresses, when they tend to increase to a fixed value of 2.7 MPa and 0.5 MPa, respectively, after 3 days. This means that the CSIBS will not decrease at high temperatures for a long time. As an important component in the cement slurry system, retarder H40L can delay the strength development of the cement. The cement strength will reach a maximum value after curing for 3 days. Moreover, the CSIBS shows the same changing trend.

## 3.4. Temperature variation

The CSIBS is usually accompanied by complicated temperature variation during well completions, including well washing, perforation, staged fracturing, and production tests. To simulate the effect of temperature variation on the CSIBS with different temperature conditions in HTHP wells, three experimental schemes of temperature variation are introduced. Figure 8 shows the experimental results when the experimental temperature increases to different values and then recovers to 150°C, proceeding for 10 minutes. The experimental results shown in figure 9 are obtained under the circumstances with the temperature decreasing from 200°C to different values. Comprehensively considering the different completion test processes usually accompanied with complex variations in temperature and pressure, figure 10 shows the complex temperature variations and their effect on the CSIBS.

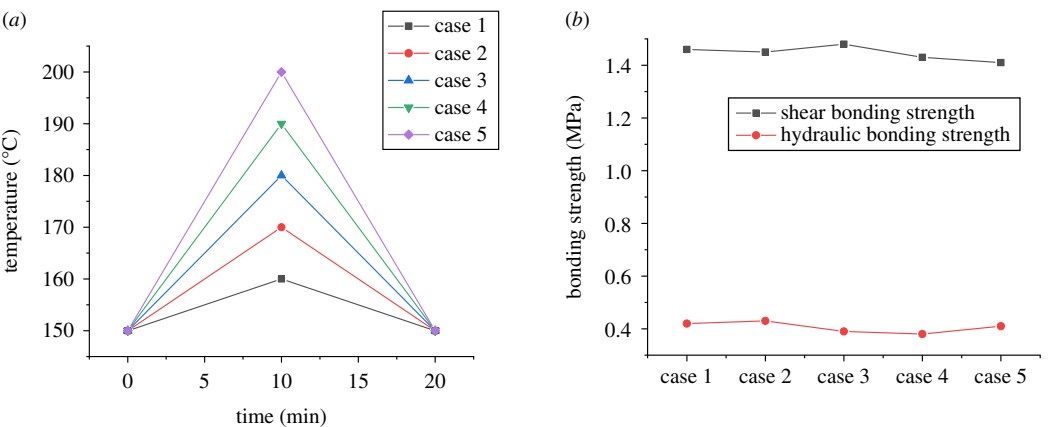

**Figure 8.** Experimental results of the CSIBS with the temperature recovering to 150℃ after increasing to different values. (*a*) Temperature variation schemes (*b*) Experimental results of the CSIBS.

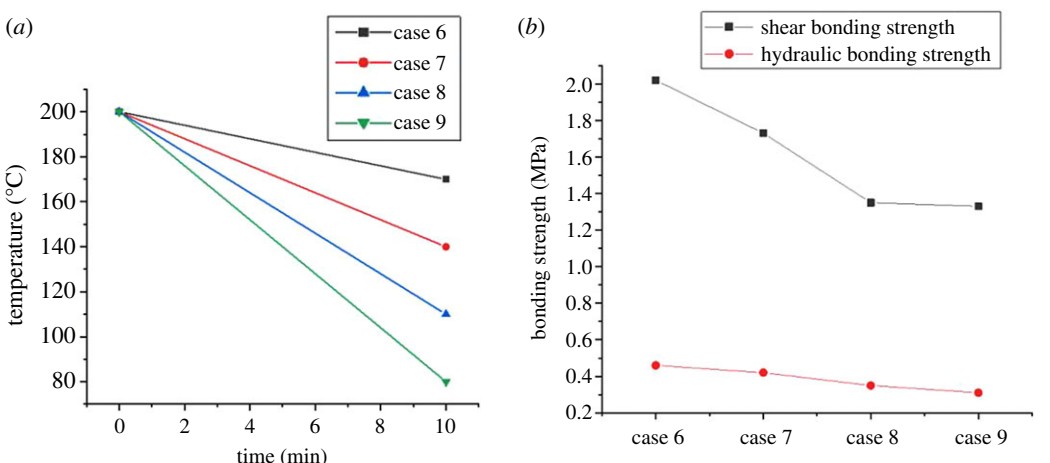

**Figure 9.** Experimental results of the CSIBS with the temperature decreasing from 200℃ to different values. (*a*) Temperature variation schemes and (*b*) experimental results of the CSIBS.

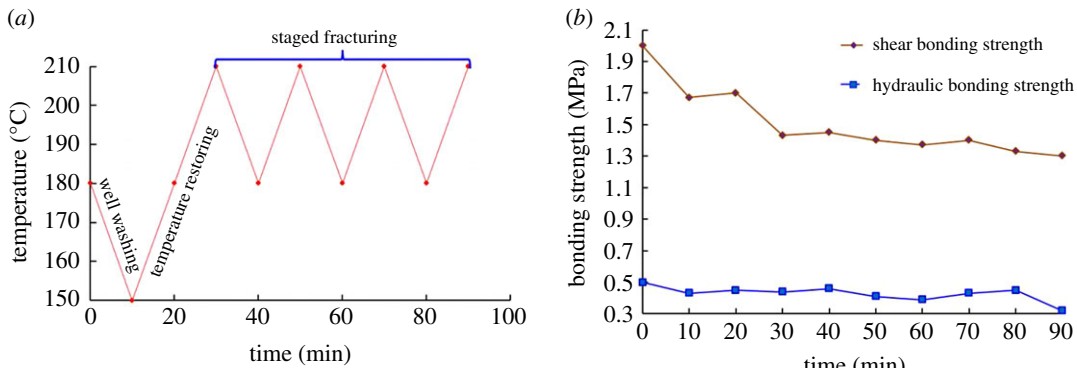

**Figure 10.** Experimental results of the CSIBS with complex variations in temperature. (*a*) Temperature variation scheme used to mimic field conditions and (*b*) experimental results of the CSIBS.

The results of the CSIBS in figures 8 and 9 indicate that the CSIBS decreases obviously due to the shrinkage of casing and cement with the reduction in the temperature. The comparison between this work and experimental data presented in the literature [15,18] indicates that the same change law is

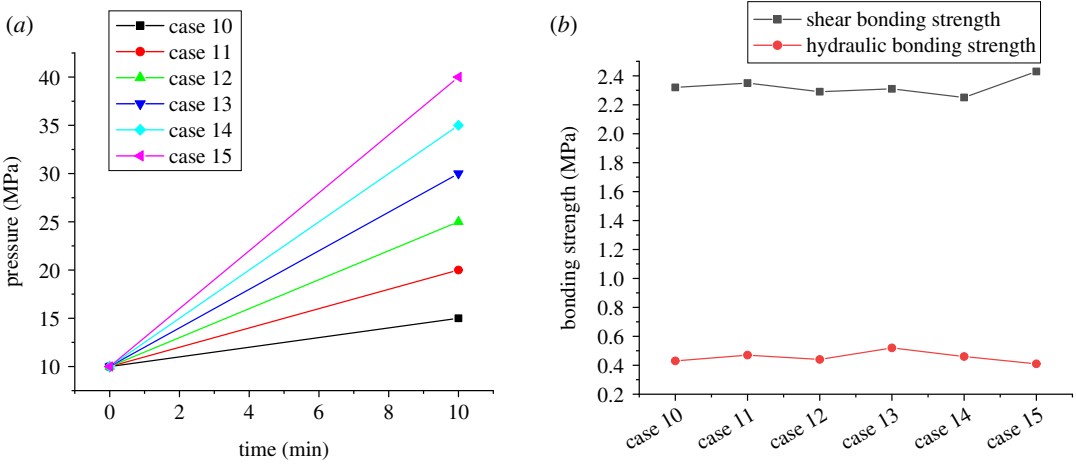

**Figure 11.** Experimental results of the CSIBS with the casing pressure increasing from 10 MPa to different values. (*a*) Casing pressure variation schemes and (*b*) experimental results of the CSIBS.

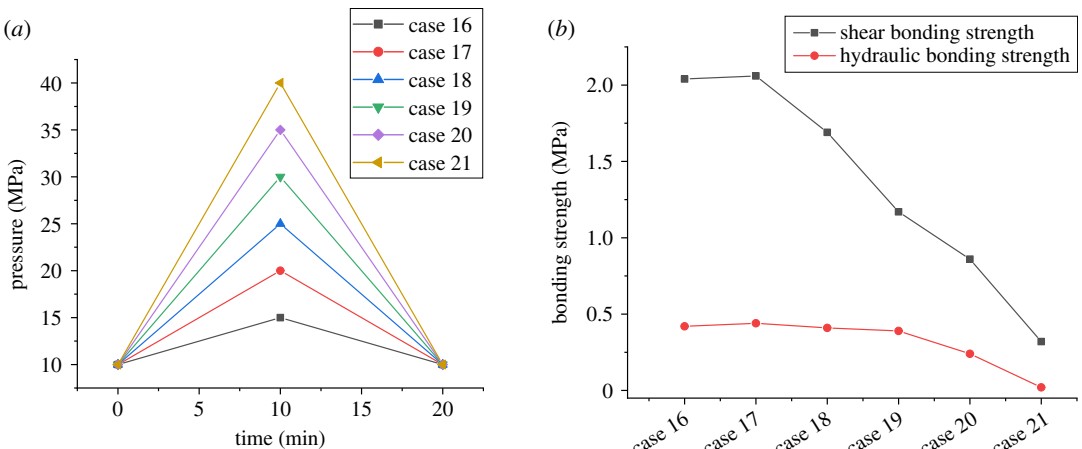

**Figure 12.** Experimental results of the CSIBS with the casing pressure recovering to 10 MPa after increasing to different values. (*a*) Casing pressure variation schemes and (*b*) experimental results of the CSIBS.

followed between the CSIBS and decreasing temperature. However, influenced by the different compositions of the cement slurry system and different properties of steel, the results do not perfectly match the previous findings even with the same temperature variation. While the temperature recovers to the initial value, the CSIBS will recover to the initial value because of the initial state of the casing and cement restoration. With the complex variation in temperature, as figure 10 shows, the shear bonding strength first decreases and then remains at a certain level, whereas the hydraulic bonding strength undergoes minor changes for the same case.

## 3.5. Internal casing pressure variation

There is usually a complex change in the casing pressure during wellbore operations, such as the casing pressure test, density reduction of the operating fluid in the casing and temperature increasing or decreasing after cementing in the HTHP wells. Pressure variations in the casing may lead to the change in stress in the casing–cement-formation system, which is the main cause of interfacial failure between casing and cement sheath. Aiming at this, four cases of variations in internal casing pressure considered in the experiments are carried out. Figures 11 and 12 show the corresponding experimental results with the internal casing pressure increasing from 10 MPa and decreasing from 20 MPa to different values, which were used to simulate the process of the casing pressure test and formation

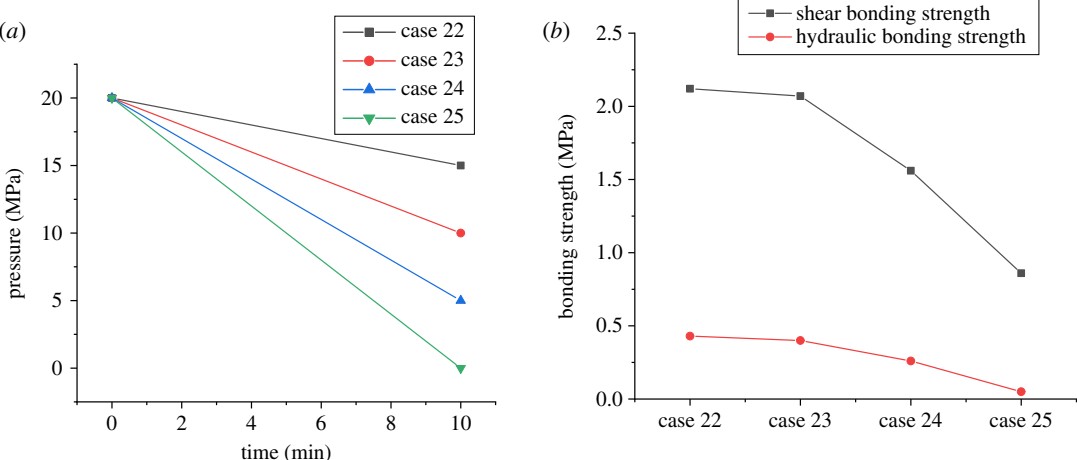

**Figure 13.** Experimental results of the CSIBS with the casing pressure decreasing from 20 MPa to different values. (*a*) Casing pressure variation schemes and (*b*) experimental results of the CSIBS.

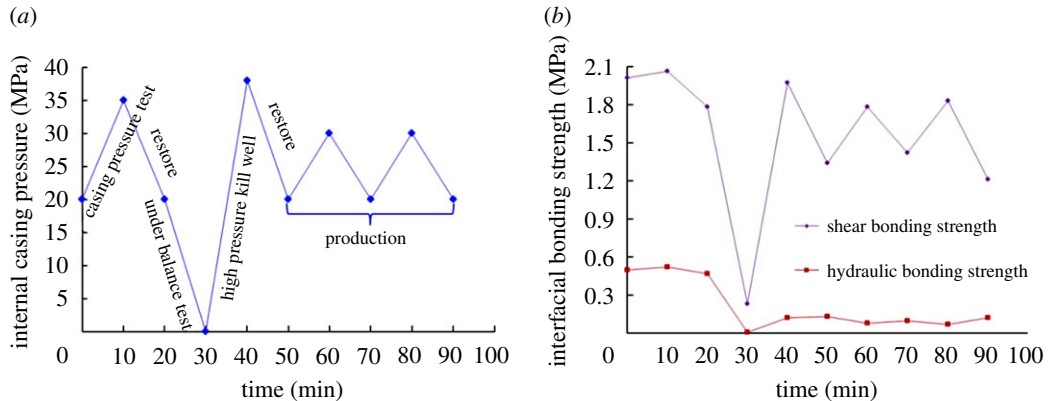

**Figure 14.** Experimental results of the CSIBS with complex variations in internal casing pressure. (*a*) Different completion process with pressure variations and (*b*) experimental results of the CSIBS.

testing. The experimental results displayed in figure 13 are obtained under the circumstances with the internal casing pressure first increasing from 10 MPa to different values over 10 minutes and then recovering to 10 MPa during the subsequent 10 minutes, which simulates the process of pressure recovery after the casing pressure test. Figure 14 shows the effect of complex variations in the internal casing pressure during the different well completion processes on CSIBS.

From figures 11–13, it can be noted that a minor change in the shear and hydraulic bonding strength occurred when the internal casing pressure increased slightly, which suggests that the strength of the cement sheath and interface can resist the load applied in the casing. However, when the increasing pressure is restored to the initial value, there exists a critical value of casing pressure variation (25 MPa in the experiment). The shear and hydraulic bonding strength will maintain values of approximately 2 and 0.42 MPa, respectively, when the pressure variation is lower than the critical value. Nevertheless, the value of the CSIBS will decrease until a minimum of approximately 0 MPa, which means that plastic deformation is generated on the cement sheath. For the case of the internal casing pressure decreasing, the CSIBS will decrease until the cementation interface of the casing and cement sheath undergo debonding because of casing shrinkage due to the decrease in casing pressure. The results of the CSIBS in figure 14 indicate that the change in the shear bonding strength is in agreement with the variations in the internal casing pressure, while the hydraulic bonding strength is difficult to restore when it decreases to a certain value. This means that the interfacial shear bond is essential to support the casing mechanically, whereas the hydraulic bond prevents the formation of micro-annuli.

# 4. Conclusion

To explore the interfacial properties of the casing and cement sheath under complex temperature and pressure conditions in HTHP wells, an experimental device was developed to measure the CSIBS, and a series of experiments were carried out. From the study and the results of the experiments, we can draw the following conclusions.

(1) The casing roughness has a significant effect on the shear bonding strength. The simulated casing with the same roughness as the actual casing should be selected first when developing the CSIBS test plan.

(2) With the curing temperature of the cement slurry increasing, the CSIBS increases and then converges to a constant value when approaching a certain curing time; thus, wellbore operations should be carried out after cementing.

(3) When the temperature or pressure inside the casing increases monotonically within a certain range, it will not cause a dramatic change in the CSIBS, but the reduction in temperature or pressure will have a large impact on the CSIBS.

(4) The CSIBS fluctuated within a narrow range when the temperature inside the casing increased first and was restored to the initial value; however, when the pressure inside changed in agreement with the same trend of the temperature, the value of the CSIBS decreased significantly. This illustrated that the recovery of the temperature has little influence on the cement ring, while the alternating change in the pressure inside the casing is much easier to cause the failure of the cement ring.

Ethics. The cement slurry is offered by CNOOC China Limited Zhanjiang Branch and is permitted to test its bonding properties in this work. The experimental device is developed by the research group which is constituted by the authors H.Y., Q.F., J.W., L.Q., D.X. and Y.L. listed in the paper.

Data accessibility. Data are available at the Dryad Digital Repository: https://doi.org/10.5061/dryad.9s4mw6mbd [24].

Authors' contributions. H.Y. designed the experimental device, carried out the experimental scheme, participated in the experiments, designed the study and drafted the manuscript; Q.F., L.Q., D.X. and Y.L. participated in the experiments; J.W. provided information on the oil and gas well completion processes and participated in the design of the experimental scheme. All authors gave final approval for publication.

Competing interests. We declare we have no competing interests.

Funding. This project is financially supported by the National Natural Science Foundation of China (grant no. 51704037).

Acknowledgements. We thank Saz Ahmed for providing helpful comments on an earlier version of the manuscript. The authors highly appreciate the anonymous reviewers for their constructive comments and suggestions.

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
