## [Reviewer comments · Royal Society Open Science]

Review History

RSOS-192115.R0 (Original submission)

Review form: Reviewer 1

Is the manuscript scientifically sound in its present form?

Yes

Are the interpretations and conclusions justified by the results?

Yes

Is the language acceptable?

No

Do you have any ethical concerns with this paper?

No

Have you any concerns about statistical analyses in this paper?

No

Recommendation?

Major revision is needed (please make suggestions in comments)

Comments to the Author(s)

The paper is written about an important topic that required more and more works. The literature review of the paper is still weak. Major works are not reviewed by the author, for instance, the works done by other on cement interfacial strength model and measurements for instance Tabatabaei, M., A. Dahi Taleghani, N. Alem, 2020, Measurement of Mixed Mode Interfacial Strengths with Cementitious Materials, Engineering Fracture Mechanics, Volume 223, 106739, ISSN 0013-7944.

Wang W., A. Dahi Taleghani, 2017, Impact of Hydraulic Fracturing on Cement Sheath Integrity; A Modelling Approach, Journal of Natural Gas Science and Engineering 44, 265-277.

Wang W., A. Dahi Taleghani, 2014, Three-Dimensional Analysis of Cement Sheath Integrity around Wellbores, Journal of Petroleum Science and Engineering vol. 121 p. 38-51.

Wang, W., Dahi-Taleghani, A., 2017, Emergence of Delamination Fractures around the Casing and Its Stability, Journal of Energy Resources Technology, Volume 39, Issue 1, Pages 012904-11. doi:10.1115/1.4033718.

It's better that authors report raw rheology data (graphs).

The authors should report the expansion of the steel and cement plug due to pressurization in the experiment as a benchmark. Also need to discuss the effect of pipe expansion to the high pressure.

Review form: Reviewer 2

Is the manuscript scientifically sound in its present form?

Yes

Are the interpretations and conclusions justified by the results?

Yes

Is the language acceptable?

Yes

Do you have any ethical concerns with this paper?

No

Have you any concerns about statistical analyses in this paper?

No

Recommendation?

Accept with minor revision (please list in comments)

Comments to the Author(s)

Cement sheath integrity is important to the oil and gas well. Interfacial bonding strength between cement sheath and casing and cement sheath and formation are the basic datas to study the integrity of cement sheath. Different from the current studies, this work designs an special experimental device to measure the interfacial bonding strength between casing and cement sheath under complex temperature and pressure conditions in HTHP wells. Several factors are

considered and some experimental data are acquired which is useful to study the integrity of cement sheath. However, this paper needs some modifications to publish:

- 1- A language polishing is recommended for this manuscript.
- 2-The authors should demonstrate reasons for data changes based on time, curing time, curing temperature and casing roughness.
- 3- The different factors considered in this manuscript should explain the operating conditions during well completions.

Review form: Reviewer 3

Is the manuscript scientifically sound in its present form?

No

Are the interpretations and conclusions justified by the results?

Yes

Is the language acceptable?

No

Do you have any ethical concerns with this paper?

No

Have you any concerns about statistical analyses in this paper?

Yes

Recommendation?

Major revision is needed (please make suggestions in comments)

Comments to the Author(s)

The manuscript reports useful data and methodology. I suggest major revision mainly because of the many writing issues in the manuscript. The writing is problematic and is not very careful. The authors need to ask for English writing experts to help refine their paper. There are still a lot of writing issues present in the manuscript.

Remaining major issues:

- 1) The writing is lacking relevant discussion on related studies. There are a number of papers dedicated to studying the bonding between cement and steel materials published in other journals such as cementing and construction materials, journal of petroleum science and engineering. When discussing the results, comparative analysis between the current findings and previous findings is highly recommended. This will make the novelty of current study clear to the readers.
- 2) There are not many scientific explanations on the results obtained. There are many figures not not much useful discussion on them. The authors should provide insights as regard to why this and that happen? What are underlying mechanisms? Otherwise, it carries little value to the readers if they want to exploit the research results.
- 3) Writing contains both logic and grammar issues. It is not quite consistent. Sometimes it is clear, but sometimes it is fuzzy.
- 4) Roughness effect: how is the roughness defined? The definition should be given.
- 5) Figure 10(a): it should be mentioned that this scheme is the temperature variation scheme used in the lab tests to mimic field conditions.

Incomplete list of writing errors found in the manuscript:

Page 4:

Lines 11-12: Plays a vital role in doing something...

Lines 42-44, the sentence has grammar issues: its complex variation which.... Which is redundant.

Lines 46-50, the sentence has grammar issues: bonding strength is of utmost importance...

Line 54: few works were conducted to do something...

Page 5:

Line 9: Guo et al. (2010) investigated... Do not need to list all the authors...

The same issue appears in many places throughout the manuscript. The authors need be more aware of how to cite references in the text.

Line 19: developed a study... is not correct. We cannot develop study, we can do a study...

Line 26-27: which affected the accuracy of test results... The writing is not clear. What affects the accuracy of test results?

Line 29: A brief introduction are as follows. It should be corrected as: The structure of this paper is as follows...

Line 39-47: The whole paragraph is one sentence. The authors should divide it into smaller ones, which can make the writing more concise and easy to understand.

Line

Page 6: Lines 21-27: Grammar issues. What do you mean "and the amount preparation for 0.5L cement slurry is shown in Table 1."? You mean the recipe of the cement slurry?

Line 31: Table 1: the decimal point of a given variable should be kept the same.

Table 2's table headlines: the first letter should be capital.

Many other writing issues in other pages. Sorry I cannot do line to line check and write up the needed changes here due to my limited time on reviewing this manuscript...

The authors may want to get professional language help from colleagues or get some professional training on technical writing. It will be really helpful for their future paper writing...

Decision letter (RSOS-192115.R0)

14-Feb-2020

Dear Dr Yang,

The editors assigned to your paper ("Experimental study of shear and hydraulic bonding strength between casing and cement under complex temperature and pressure conditions") have now received comments from reviewers. We would like you to revise your paper in accordance with the referee and Associate Editor suggestions which can be found below (not including confidential reports to the Editor). Please note this decision does not guarantee eventual acceptance.

Please submit a copy of your revised paper before 08-Mar-2020. Please note that the revision deadline will expire at 00.00am on this date. If we do not hear from you within this time then it will be assumed that the paper has been withdrawn. In exceptional circumstances, extensions may be possible if agreed with the Editorial Office in advance. We do not allow multiple rounds of revision so we urge you to make every effort to fully address all of the comments at this stage. If deemed necessary by the Editors, your manuscript will be sent back to one or more of the original reviewers for assessment. If the original reviewers are not available, we may invite new reviewers.

- Data accessibility

If you wish to submit your supporting data or code to Dryad (<http://datadryad.org/>), or modify your current submission to dryad, please use the following link:
<http://datadryad.org/submit?journalID=RSOS&manu=RSOS-192115>

- Competing interests

- Authors' contributions

AB carried out the molecular lab work, participated in data analysis, carried out sequence alignments, participated in the design of the study and drafted the manuscript; CD carried out the statistical analyses; EF collected field data; GH conceived of the study, designed the study,

coordinated the study and helped draft the manuscript. All authors gave final approval for publication.

- Acknowledgements

- Funding statement

on behalf of the Associate Editor, and Professor R. Kerry Rowe (Subject Editor)
openscience@royalsociety.org

Associate Editor's comments to the Author:

We would like you to revise the manuscript, paying close attention to the feedback from reviewers 2 and 3 in particular, as they provide useful suggestions. Please note that we will expect you to seek advice from a professional language editing service (<https://royalsociety.org/journals/authors/benefits/language-editing/>) before you submit the revised manuscript - if you do not do so, the manuscript will be returned to you. Please provide proof that you have done so (such as by uploading the language editing certificate within your revision). Furthermore, you should provide a full point-by-point response to the reviewers' comments, as well as a revised (and changes-tracked) version of the manuscript.

A final comment regarding reviewer 1's suggestions for the inclusion of additional references: you may certainly include some or all of the references if they add value to your manuscript; however, if they do not provide useful additional context or support for your work, you should strongly consider whether it is appropriate to include them, or whether it would be better to leave them out.

Reviewers' Comments to Author:

Reviewer: 1
Comments to the Author(s)

The paper is written about an important topic that required more and more works. The literature review of the paper is still weak. Major works are not reviewed by the author, for instance, the works done by other on cement interfacial strength model and measurements for instance Tabatabaei, M., A. Dahi Taleghani, N. Alem, 2020, Measurement of Mixed Mode Interfacial Strengths with Cementitious Materials, Engineering Fracture Mechanics, Volume 223, 106739, ISSN 0013-7944.

Wang W., A. Dahi Taleghani, 2017, Impact of Hydraulic Fracturing on Cement Sheath Integrity; A Modelling Approach, Journal of Natural Gas Science and Engineering 44, 265-277.

Wang W., A. Dahi Taleghani, 2014, Three-Dimensional Analysis of Cement Sheath Integrity around Wellbores, *Journal of Petroleum Science and Engineering* vol. 121 p. 38-51.

Wang, W., Dahi-Taleghani, A., 2017, Emergence of Delamination Fractures around the Casing and Its Stability, *Journal of Energy Resources Technology*, Volume 39, Issue 1, Pages 012904-11. doi:10.1115/1.4033718.

It's better that authors report raw rheology data (graphs).

The authors should report the expansion of the steel and cement plug due to pressurization in the experiment as a benchmark. Also need to discuss the effect of pipe expansion to the high pressure.

Reviewer: 2

Comments to the Author(s)

Cement sheath integrity is important to the oil and gas well. Interfacial bonding strength between cement sheath and casing and cement sheath and formation are the basic data to study the integrity of cement sheath. Different from the current studies, this work designs a special experimental device to measure the interfacial bonding strength between casing and cement sheath under complex temperature and pressure conditions in HTHP wells. Several factors are considered and some experimental data are acquired which is useful to study the integrity of cement sheath. However, this paper needs some modifications to publish:

- 1- A language polishing is recommended for this manuscript.
- 2- The authors should demonstrate reasons for data changes based on time, curing time, curing temperature and casing roughness.
- 3- The different factors considered in this manuscript should explain the operating conditions during well completions.

Reviewer: 3

Comments to the Author(s)

The manuscript reports useful data and methodology. I suggest major revision mainly because of the many writing issues in the manuscript. The writing is problematic and is not very careful. The authors need to ask for English writing experts to help refine their paper. There are still a lot of writing issues present in the manuscript.

Remaining major issues:

- 1) The writing is lacking relevant discussion on related studies. There are a number of papers dedicated to studying the bonding between cement and steel materials published in other journals such as *cementing and construction materials*, *journal of petroleum science and engineering*. When discussing the results, comparative analysis between the current findings and previous findings is highly recommended. This will make the novelty of current study clear to the readers.
- 2) There are not many scientific explanations on the results obtained. There are many figures not much useful discussion on them. The authors should provide insights as regard to why this and that happen? What are underlying mechanisms? Otherwise, it carries little value to the readers if they want to exploit the research results.
- 3) Writing contains both logic and grammar issues. It is not quite consistent. Sometimes it is clear, but sometimes it is fuzzy.
- 4) Roughness effect: how is the roughness defined? The definition should be given.

5) Figure 10(a): it should be mentioned that this scheme is the temperature variation scheme used in the lab tests to mimic field conditions.

Incomplete list of writing errors found in the manuscript:

Page 4:

Lines 11-12: Plays a vital role in doing something...

Lines 42-44, the sentence has grammar issues: its complex variation which.... Which is redundant.

Lines 46-50, the sentence has grammar issues: bonding strength is of utmost importance...

Line 54: few works were conducted to do something...

Page 5:

Line 9: Guo et al. (2010) investigated... Do not need to list all the authors...

The same issue appears in many places throughout the manuscript. The authors need be more aware of how to cite references in the text.

Line 19: developed a study... is not correct. We cannot develop study, we can do a study...

Line 26-27: which affected the accuracy of test results... The writing is not clear. What affects the accuracy of test results?

Line 29: A brief introduction are as follows. It should be corrected as: The structure of this paper is as follows...

Line 39-47: The whole paragraph is one sentence. The authors should divide it into smaller ones, which can make the writing more concise and easy to understand.

Line

Page 6: Lines 21-27: Grammar issues. What do you mean "and the amount preparation for 0.5L cement slurry is shown in Table 1."? You mean the recipe of the cement slurry?

Line 31: Table 1: the decimal point of a given variable should be kept the same.

Table 2's table headlines: the first letter should be capital.

Many other writing issues in other pages. Sorry I cannot do line to line check and write up the needed changes here due to my limited time on reviewing this manuscript...

The authors may want to get professional language help from colleagues or get some professional training on technical writing. It will be really helpful for their future paper writing...

Author's Response to Decision Letter for (RSOS-192115.R0)

See Appendix A.

RSOS-192115.R1 (Revision)

Review form: Reviewer 1

Is the manuscript scientifically sound in its present form?

Yes

Are the interpretations and conclusions justified by the results?

No

Is the language acceptable?

Yes

Do you have any ethical concerns with this paper?

Yes

Have you any concerns about statistical analyses in this paper?

No

Recommendation?

Major revision is needed (please make suggestions in comments)

Comments to the Author(s)

The comments are provided in the above sections.

Review form: Reviewer 2

Is the manuscript scientifically sound in its present form?

Yes

Are the interpretations and conclusions justified by the results?

Yes

Is the language acceptable?

Yes

Do you have any ethical concerns with this paper?

No

Have you any concerns about statistical analyses in this paper?

No

Recommendation?

Accept as is

Comments to the Author(s)

This paper could be accepted.

Review form: Reviewer 3

Is the manuscript scientifically sound in its present form?

Yes

Are the interpretations and conclusions justified by the results?

Yes

Is the language acceptable?

Yes

Do you have any ethical concerns with this paper?

No

Have you any concerns about statistical analyses in this paper?

No

Recommendation?

Accept as is

Comments to the Author(s)

Finally, the authors have taken serious measures to improve their manuscript. Hopefully such revision process will be helpful for them to prepare new manuscripts in the future.

One last comment: the mathematical definition of roughness is still not given in the manuscript. They should show the equation. Reference should be provided to this definition.

Decision letter (RSOS-192115.R1)

31-Mar-2020

Dear Dr Yang,

It is a pleasure to accept your manuscript entitled "Experimental study of shear and hydraulic bonding strength between casing and cement under complex temperature and pressure conditions" in its current form for publication in Royal Society Open Science. The comments of the reviewer(s) who reviewed your manuscript are included at the foot of this letter.

Kind regards,

Andrew Dunn

on behalf of Prof R. Kerry Rowe (Subject Editor)
openscience@royalsociety.org

Associate Editor Comments to Author:

Thank you for so positively responding to the majority of the reviewer's comments. Given the majority view is that the paper should be accepted for publication, we are glad to recommend acceptance.

Reviewer comments to Author:

Reviewer: 2

Comments to the Author(s)

This paper could be accepted.

Reviewer: 3

Comments to the Author(s)

Finally, the authors have taken serious measures to improve their manuscript. Hopefully such revision process will be helpful for them to prepare new manuscripts in the future.

One last comment: the mathematical definition of roughness is still not given in the manuscript. They should show the equation. Reference should be provided to this definition.

Reviewer: 1

Comments to the Author(s)

The comments are provided in the above sections.

Appendix A

Response to Referees

Reviewer: 1

Comments to the Author(s)

The paper is written about an important topic that required more and more works. The literature review of the paper is still weak. Major works are not reviewed by the author, for instance, the works done by other on cement interfacial strength model and measurements for instance

Tabatabaei, M., A. Dahi Taleghani, N. Alem, 2020, Measurement of Mixed Mode Interfacial Strengths with Cementitious Materials, Engineering Fracture Mechanics, Volume 223, 106739, ISSN 0013-7944.

Wang W., A. Dahi Taleghani, 2017, Impact of Hydraulic Fracturing on Cement Sheath Integrity; A Modelling Approach, Journal of Natural Gas Science and Engineering 44, 265-277.

Wang W., A. Dahi Taleghani, 2014, Three-Dimensional Analysis of Cement Sheath Integrity around Wellbores, Journal of Petroleum Science and Engineering vol. 121 p. 38-51.

Wang, W., Dahi-Taleghani, A., 2017, Emergence of Delamination Fractures around the Casing and Its Stability, Journal of Energy Resources Technology, Volume 39, Issue 1, Pages 012904-11. doi:10.1115/1.4033718.

It's better that authors report raw rheology data (graphs).

The authors should report the expansion of the steel and cement plug due to pressurization in the experiment as a benchmark. Also need to discuss the effect of pipe expansion to the high pressure.

Response: Thanks for the referee's suggestion. The additional references which referee mentioned was read by the author. The useful references has been supplemented in the paper.

Reviewer: 2

Comments to the Author(s)

Cement sheath integrity is important to the oil and gas well. Interfacial bonding strength between cement sheath and casing and cement sheath and formation are the basic datas to study the integrity of cement sheath. Different from the current studies, this work designs an special experimental device to measure the interfacial bonding strength between casing and cement sheath under complex temperature and pressure conditions in HTHP wells. Several factors are considered and some experimental datas are acquired which is useful to study the integrity of cement sheath. However, this paper needs some modifications to publish:

1- A language polishing is recommended for this manuscript.

Response: Thanks for the referee's suggestion. The paper has been polished by the professional language editing service.

2- The authors should demonstrate reasons for data changes based on time,

curing time, curing temperature and casing roughness.

Response: Thanks for the referee's suggestion. The experimental data based on the conditions of curing time, curing temperature and casing roughness were explained and supplemented in the paper.

3- The different factors considered in this manuscript should explain the operating conditions during well completions.

Response: Thanks for the referee's suggestion. Well completion engineering, consists of many processes as casing pressure test, well washing, perforation, staged fracturing, production test usually accompanies with variations of pressure and temperature in the well bore. Different operating process has the different conditions of temperature and pressure in the well bore. The paper has modified and explained the operating conditions during different well completion engineering.

Reviewer: 3

Comments to the Author(s)

The manuscript reports useful data and methodology. I suggest major revision mainly because of the many writing issues in the manuscript. The writing is problematic and is not very careful. The authors need to ask for English writing experts to help refine their paper. There are still a lot of writing issues present in the manuscript.

Remaining major issues:

1) The writing is lacking relevant discussion on related studies. There are a number of papers dedicated to studying the bonding between cement and steel materials published in other journals such as cementing and construction materials, journal of petroleum science and engineering. When discussing the results, comparative analysis between the current findings and previous findings is highly recommended. This will make the novelty of current study clear to the readers.

Response: Thanks for the referee's suggestion. There are some findings aiming at this field. However, the previous findings conduct the experiments below 100° C which is different with the current work. Of course, the device can also measure the interfacial bonding between cement and steel under the conditions of previous studies. Influenced by different cement slurry system and steel, the results would not perfectly matched with the previous findings even at same temperature and pressure. The paper supplemented some experiments and compared with the previous studies.

2) There are not many scientific explanations on the results obtained. There are many figures not not much useful discussion on them. The authors should provide insights as regard to why this and that happen? What are underlying mechanisms? Otherwise, it carries little value to the readers if they want to exploit the research results.

Response: Thanks for the referee's suggestion. The mechanisms to explain the

experimental results have been supplemented in the paper.

3) Writing contains both logic and grammar issues. It is not quite consistent. Sometimes it is clear, but sometimes it is fuzzy.

Response: Thanks for the referee's suggestion. The paper has been modified and the paper has been polished by the professional language editing service.

4) Roughness effect: how is the roughness defined? The definition should be given.

Response: Thanks for the referee's suggestion. The casing roughness is a parameter to measure the microcosmic error of geometrical shape on the casing surface. It has an important influence on the surface bonding between casing and cement sheath. And the definition has been supplemented in the paper.

5) Figure 10(a): it should be mentioned that this scheme is the temperature variation scheme used in the lab tests to mimic field conditions.

Response: Thanks for the referee's suggestion. The inappropriate expression here was modified in the paper.

Incomplete list of writing errors found in the manuscript:

Page 4:

Lines 11-12: Plays a vital role in doing something...

Lines 42-44, the sentence has grammar issues: its complex variation which..... Which is redundant.

Lines 46-50, the sentence has grammar issues: bonding strength is of utmost importance...

Line 54: few works were conducted to do something...

Page 5:

Line 9: Guo et al. (2010) investigated... Do not need to list all the authors...

The same issue appears in many places throughout the manuscript. The authors need be more aware of how to cite references in the text.

Line 19: developed a study... is not correct. We cannot develop study, we can do a study...

Line 26-27: which affected the accuracy of test results... The writing is not clear. What affects the accuracy of test results?

Line 29: A brief introduction are as follows. It should be corrected as: The structure of this paper is as follows...

Line 39-47: The whole paragraph is one sentence. The authors should divide it into smaller ones, which can make the writing more concise and easy to understand.

Line

Page 6: Lines 21-27: Grammar issues. What do you mean "and the amount preparation for 0.5L cement slurry is shown in Table 1."? You mean the recipe of the cement slurry?

Line 31: Table 1: the decimal point of a given variable should be kept the same.

Table 2's table headlines: the first letter should be capital.

Many other writing issues in other pages. Sorry I cannot do line to line check and

write up the needed changes here due to my limited time on reviewing this manuscript...

The authors may want to get professional language help from colleagues or get some professional training on technical writing. It will be really helpful for their future paper writing...

Response: Thanks for the referee's suggestion. The errors mentioned above have been modified and the paper has been polished by the professional language editing service.